# Tissue of origin characterization of cell free DNA in seminal plasma: Implications for new liquid biopsies

**Stephanie Huang**[1]☯, **James C. Hart**[1]☯, **James F. Smith**[1,2], **Shellie Bench**[1], **Laura Rivas Yepes**[1], **Bailey Griscom**[1], **Kim M. Clark-Langone**[1]*

1 Fellow Health Inc, San Leandro, California, United States of America, 2 Department of Urology, University of California, San Francisco, San Francisco, California, United States of America

☯ These authors contributed equally to this work.
* kclangone@meetfellow.com

## Abstract

Liquid biopsies are becoming increasingly used for the detection and monitoring of disease states. While cell free DNA (cfDNA) in blood and urine have been well studied, much less is known about the composition of cfDNA in seminal fluid. We sought to characterize cfDNA in seminal fluid through tissue of origin studies using methylation analysis in men aged 21–60 yrs. We confirmed the observations of others that seminal fluid contains an abundance of cfDNA that is both nucleosomal and > 1 kb. However, here we demonstrate for the first time that the high molecular weight (HMW) DNA harbors a lower sperm signal and higher somatic cell signal compared to the nucleosomal fraction. Prostate, granulocytes and kidney showed a mean predicted increased contribution of 6.2%, 4.9% and 2.9%, respectively in the HMW fraction. While sperm was the predominant signal in most men without vasectomies, granulocyte cfDNA made up most of the signal in two of the non-vasectomy subjects. Unexpectedly, the proportion of prostate signal reached as high as 26.5% in the HMW fraction in non-vasectomy subjects. We also observed subject-specific cfDNA size distribution patterns that were reproducible over time, irrespective of abstinence times. These results suggest that seminal fluid is a rich source of cfDNA from various somatic cell types, and enriching for the HMW fraction would yield even higher sensitivity for somatic cfDNA detection. Considering these novel findings, it appears that seminal fluid may be able to serve as liquid biopsy for the detection and monitoring of prostate cancer, benign prostate hyperplasia, prostatitis and infertility.

## Introduction

Seminal fluid is a complex mixture of secretions from the seminal gland, prostate, testes, epididymis, and bulbourethral glands. It not only serves as a medium for sperm transport but also provides nutrients and a protective environment for spermatozoa during their journey through the female reproductive tract. Seminal fluid contains a complex range of organic and inorganic constituents, including proteins, enzymes, lipids, carbohydrates, and various ions, which play crucial roles in sperm motility and viability. Despite the emerging interest in liquid

**Data availability statement:** The data for this study are publicly available from the NCBI BioProject repository (https://www.ncbi.nlm.nih.gov/bioproject/PRJNA1162531).

**Funding:** This study was funded by Fellow Health Inc., a private company funded by various Venture Capital funds. The funders had no role in study design, data collection and analysis, decision to publish, or preparation of the manuscript.

**Competing interests:** I have read the journal's policy and the authors have the following competing interests: S.H., J.C.H., J.F.S., S.B., L.R.Y., B.G. and K.M.C.L. are employees of Fellow Health Inc. and have stock ownership. S.H., J.C.H. and K.M.C.L. are inventors on patent applications related to cfDNA isolation from seminal plasma. This does not alter our adherence to PLOS ONE policies on sharing data and materials.

biopsies in cancer diagnostics and beyond, seminal fluid has not been extensively studied as a potential specimen type. Here we characterize the composition of cell-free DNA (cfDNA) in seminal fluid, highlighting its potential use in the liquid biopsy space.

Extracellular DNA (exDNA) or cfDNA is thought to be released into the extracellular environment through apoptosis, necrosis, NETosis, or active secretion by cells [1,2]. Given the abundance of sperm in semen, one could expect the large majority of cfDNA in seminal fluid to arise from apoptosis of sperm cells either undergoing spermatogenesis in close proximity to the seminiferous tubule lumen, or during storage of mature sperm in the epididymis. Such cfDNA could serve as biomarkers for fertility as reported by Chou et al. [3], whereby they demonstrated correlations between cfDNA size and important sperm parameters, and Di Pizio et al. [4] who found significantly higher cfDNA levels in patients with sperm abnormalities compared to controls. Other sources of cfDNA in semen could arise from the prostate, bladder, kidney, reproductive tract cells, or resident immune cells. One group proposed seminal plasma cfDNA as a potential biomarker for prostate cancer given their observation of significantly higher concentrations of cfDNA in prostate cancer compared to controls, and larger cfDNA fragment sizes in prostate cancer patients compared to those with benign prostate hyperplasia (BPH) and healthy controls [5–7]. In their study on cfDNA levels and its association with sperm abnormalities, Di Pizio et al. [4] concluded that it may be of interest to study the cfDNA's origin and clearance and its methylation profile. One group showed LGALS3 cfDNA methylation status in seminal fluid to be able to discriminate between prostate cancer and benign prostate hyperplasia [8], but to date, no-one to our knowledge has fully characterized the cfDNA methylation signatures present in seminal fluid to understand the relative abundance of cfDNA from the various cell/tissue types that make up the male genitourinary system. Moreover, to our knowledge, no-one has characterized the cell/tissue of origin of the various cfDNA fragment lengths that have previously been correlated with prostate cancer [7]. We sought to further investigate these findings by characterizing cfDNA yield, fragment size, and cell/tissue of origin from healthy volunteers. Once the composition of cfDNA in seminal fluid is better understood, we can begin to think about its potential use as a liquid biopsy for fertility, prostate cancer, testicular cancer, benign prostate hyperplasia and prostatitis.

In terms of stability, liquid biopsy test manufacturers working with blood and urine have adopted the use of various preservatives to stabilize cells and nucleic acids so samples can be shipped without fear of losing sensitivity [9,10]. If blood is collected without the use of a suitable preservative, cell free nucleic acids will degrade, and white blood cells will die and lyse releasing high molecular weight DNA. This will result in a massive amount of background signal that would likely mask the signal of the already low analyte of interest. To date however, there are no described preservatives that have been shown to stabilize sperm and nucleic acids in seminal fluid.

The objectives of this study were to characterize the tissue of origin of the various cfDNA molecular weight fractions previously identified in seminal fluid. After identifying a suitable preservative, we were able to demonstrate that seminal fluid harbors an abundance of cfDNA from somatic cells, in addition to that from sperm cells, and that the high molecular weight (HMW) DNA harbors a higher abundance of somatic cell DNA than the nucleosomal fraction.

## Materials and methods

### Clinical study information

Samples were collected from subjects enrolled into two clinical studies, with study tracking numbers 20222556 and 20234407. Each study received ethical approval by an Institutional

Review Board (IRB00000533), and written informed consent was obtained electronically from study participants prior to engaging in study activities. Participants in study 20222556 were enrolled from 15Feb2023 to 18Oct2023, and participants in study 20234407 were enrolled from 5Oct2023 to 18Oct2023. Vasectomy status was captured as part of the enrollment process.

## Seminal fluid collection

Fresh semen samples were obtained from men aged 21–60 yrs. Participants were instructed to collect seminal fluid through masturbation after a period of 2–6 days abstinence. In some cases, participants dropped the sample off at the Fellow laboratory within 2 hours of collection and seminal plasma was then prepared immediately, or preservative added to the seminal fluid sample and left for up to 3 days prior to seminal plasma preparation. In other cases, preservative was added immediately after collection (within 30 minutes) and the sample left for up to 3 days prior to seminal plasma preparation. Adding preservative immediately and waiting for 3 days before preparing seminal plasma best represents the scenario of an at home collection test. As part of a stability study, semen samples from 3 subjects were placed in a temperature-controlled chamber and the temperature cycled from 10°C to 35°C to mimic shipping conditions.

## Seminal plasma preparation

Seminal plasma was prepared by centrifuging the semen at 400 x g for 15 mins at room temperature to pellet sperm cells and somatic cells. The supernatant was transferred to a new tube and a second spin performed at 16,000 x g for 10 mins to pellet cellular debris and the protamine associated cfDNA from mature sperm. Seminal plasma was stored at −80°C prior to cfDNA extraction.

## cfDNA extraction and size selection

cfDNA was extracted from seminal plasma following Qiagen's Circulating Nucleic Acids Extraction kit instructions using 1–3 ml seminal plasma. cfDNA was then quantitated using Qubit (Thermo Fisher) and the cfDNA profile obtained from the Tapestation 2200 (Agilent). High molecular weight (HMW) cfDNA (>800 bp) and small molecular weight cfDNA (<500 bp) were separated as follows: a double size selection was performed using a 0.6x/1.8x ratio of SPRI reagent to sample volume. The HMW DNA was eluted from the beads and underwent a further size selection using 0.5x ratio of SPRI reagent to sample volume. The HMW DNA was subject to 250 seconds of sonication using a Covaris ultrasonicator, followed by a SPRI clean up using 1.5x ratio of SPRI reagent to sample volume. After size selection, there was sufficient small sized selected cfDNA (SSD) from 19 subjects (15 non-vasectomy and 4 vasectomy) to take into library prep. After size selection, sonication and clean up for the HMW DNA, there was sufficient DNA from 15 subjects (11 non-vasectomy and 4 vasectomy). All 15 HMW samples had a paired small sized selected cfDNA (SSD) sample.

## Methylation library preparation and sequencing

Methylation libraries were prepared using 7–40 ng of DNA for both small sized selected cfDNA (SSD) and HMW using New England Biolab's NEBNext® Enyzmatic Methyl-Seq kit, according to the manufacturer's instructions. Target enrichment was then performed using Twist's Human Methylome Panel, according to the manufacturer's instructions. Next Generation Sequencing was performed on a NovaSeqX (either on a 10B or 25B flowcell), 2x150bp reads, at a depth of $5x10^7$–$2.5x10^8$ reads.

## Methylation computational processing

Following demultiplexing, reads were trimmed to remove adapters and low quality sequences using fastp (v0.23.4) [11] (extra options: '--trim_poly_g -f 1'). Reads were aligned to GRCh38 using bwameth (v0.2.7) [12] and bwa mem2 (v2.2.1) [13]. Following alignment, reads were sorted and indexed using samtools (v1.3) [14], and read duplicates were marked using picard-tools v3.1.0 (Picard).CpG and CHH methylation content was tabulated using MethylDackel (v0.6.1) (MethylDackel), only within regions covered by the Twist Human Methylome Panel, using the options '--CHH --nOT 3,0,0,3 --nOB 0,3,3,0' to exclude read ends with observed decreased methyl conversion in control materials. For each CpG, the *Beta* value (methylated reads)/(methylated reads + unmethylated reads) was tabulated.

## Methylation tissue deconvolution

Reference tissue datasets were processed from previously published studies (S1 Table) [15–18]. When raw sequencing reads were available, the same methylation processing pipeline described above was used. Otherwise, the processed CpG methylated/unmethylated counts supplied by the study were used to compute *Beta* values. CpGs were grouped into regions based on the Twist Human Methylome Panel, with the overall methylation *Beta* value calculated using the median of the CpG level *Beta* values. For each tissue, 25 marker regions were selected using a one-vs-all approach detailed in Loyfer at al. [15]. In short, tissue specific hypomethylated markers were selected based on the difference between the 75th percentile within the given tissue vs the 2.5th percentile for the remaining samples in other tissues. The top 25 markers based on this score were selected, and the tissue signature methylation profile was calculated as the median region level *Beta* value for all reference samples from the given tissue, restricted to all markers selected across all reference tissues (S2 Table). As an assessment of tissue functionality, DNase I hypersensitivity peaks were downloaded from ENCODE (S3 Table), and checked for proximity to the tissue methylation markers.

A methyl region *Beta* value was tabulated for each seminal plasma sample, again using the median CpG level *Beta* value for the given sample. Using the above tissue signature methylation profiles, SciPy non-negative least squares implementation was used to find the optimal coefficients of each tissue contributing to the given sample's profile. All coefficients were normalized to sum to 1, ensuring estimations could be interpreted as proportions. Scripts needed to run the methylation tissue deconvolution can be found on GitHub (https://github.com/meetfellow/fellow-genomics).

## Results

### Stabilization of sperm/cfDNA

To enable a semen sample to be produced at home as a liquid biopsy, it would be necessary to use a preservative that stabilizes sperm in addition to maintaining the integrity of the cfDNA during shipment to the laboratory. In this study we identified a preservative that could successfully achieve this. Fig 1 illustrates a comparison of the same sample with or without preservative. Fig 1A shows that on Day 0 the profiles look largely the same (both were processed within 2 hours of collection). At Day 3 in the oven (with oscillating temperatures from 10°C-35°C), the sample with preservative maintains a very similar profile (Fig 1B), while the one without preservative shows evidence of cell lysis and a massive increase in the amount of DNA recovered (Fig 1C). Note that when cfDNA was extracted from seminal fluid without preservative but within 2 hours of collection, we usually observed nucleosomal peaks (~150–200 bp)

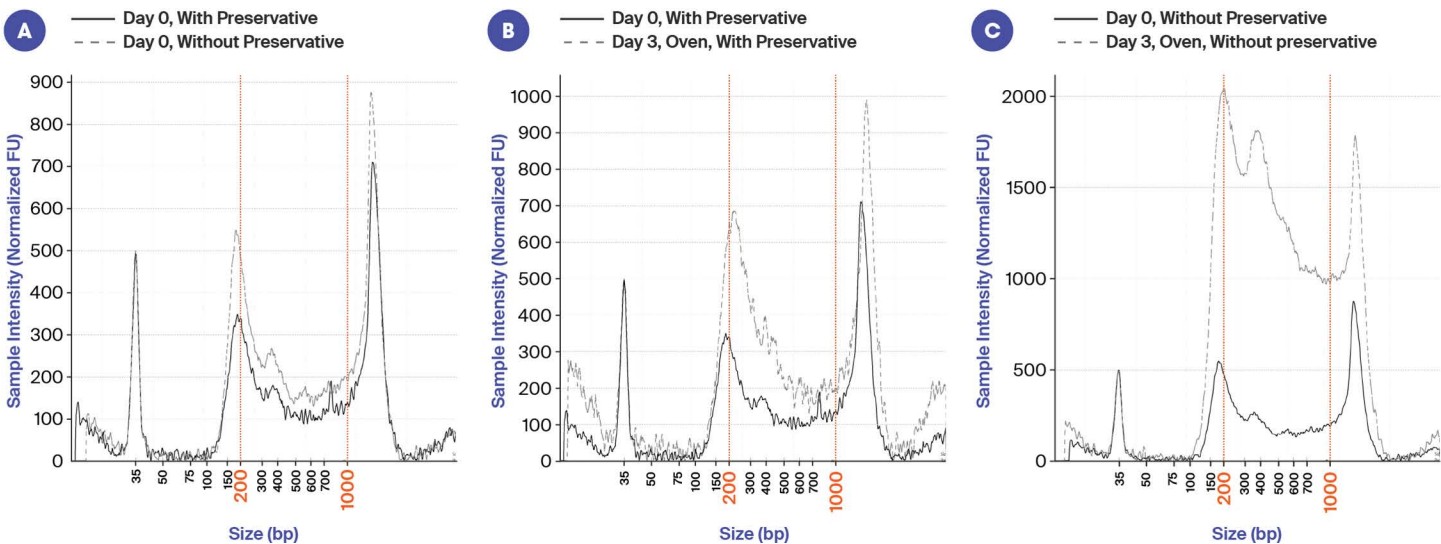

**Fig 1. Electropherogram images illustrating the stabilization of sperm and cfDNA with preservation solution. x-axis is size (bp), and y-axis is signal intensity.** A: Comparison of the same sample with and without preservative on Day 0 (seminal fluid processed within 2 hrs of collection). B: Sample with preservative on Day 0 and Day 3 in the oven. C: Sample without preservative on Day 0 and Day 3 in the oven.

in addition to a peak > 1 kb (Fig 1A). However, in some individuals, only the peak > 1 kb was observed (see cfDNA size distribution profile section below).

## cfDNA yield

Similar to other studies, we found the cfDNA yield in seminal plasma to be very high. The average total yield was 1.2 ug and there was a weak correlation between total yield and the volume of seminal plasma used in DNA extraction (Fig 2A). When normalized to the volume of seminal plasma, the average was 1003 ng per ml seminal plasma, which is at least an order of magnitude higher than that seen in blood plasma or urine. Values ranged from 260 ng–3229 ng per ml seminal plasma (Fig 2B). Not surprisingly, the vasectomy subjects tended to have lower yields of cfDNA, however they were not clear outliers (Fig 2A, B), again demonstrating that not all cfDNA in seminal fluid is sperm derived. Neither yield nor volume was highly correlated with age (S1 Fig).

## cfDNA size distribution profile by electrophoresis

In this study we observed several different characteristic cfDNA size-distribution profiles that appear to be subject specific and may be associated with interesting underlying biology. In most cases the predominant nucleosomal peak appeared to be closer to 200 bp by Tapestation analysis (see Fig 3A) rather than the typical 166 bp characteristic of nucleosomal cfDNA, but upon sequencing, the mean peak insert size was 146 bp. In some cases, there was a single clean nucleosomal peak (Fig 3A), while others had clear multi-nucleosomal peaks (Fig 3B) along with the > 1 kb peak. In the case of vasectomy participants, we observed no prominent distinct nucleosomal cfDNA peak, but only the peak > 1 kb (Fig 3C). Given the absence of an abundant distinct peak of small (nucleosomal) cfDNA in men having undergone vasectomy, this suggests that most of the small cfDNA is coming from sperm. It is worth noting that in every participant, including those with vasectomies, there was a peak > 1 kb. This was of great interest as it indicated that the cfDNA > 1 kb was derived, at least in part, from somatic cells.

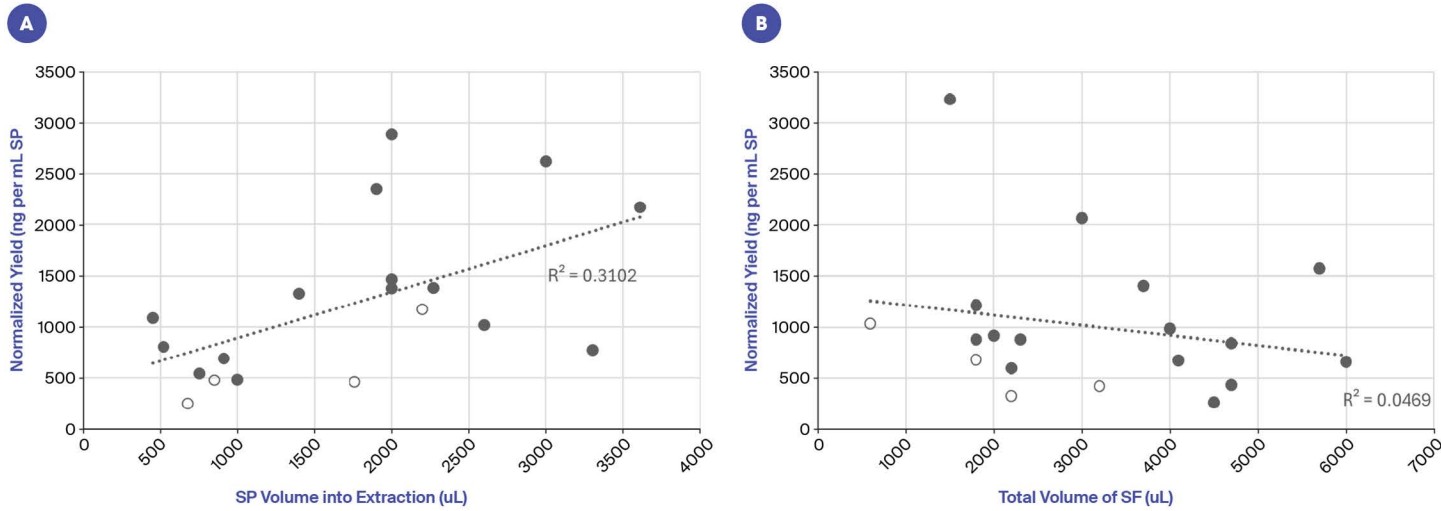

**Fig 2. cfDNA yields correlated weakly with volume.** A: Total yield (ng) by volume of seminal plasma (SP) into extraction. Solid circles are non-vasectomy subjects, open circles are vasectomy subjects. B: Normalized yield (ng per ml SP) by volume of seminal fluid (SF). Solid circles are non-vasectomy subjects, open circles are vasectomy subjects.

To investigate whether the variability between subjects could also be associated with abstinence times, two subjects provided samples after various periods of abstinence. As can be seen in Fig 3D, abstinence time did not appear to be a factor in the overall profile, but did seem to impact the total amount of cfDNA. After 6 days of abstinence the yield was 2297 ng/ml plasma as compared to only 989 ng/ml plasma after 1 day of abstinence. However, the percent nucleosomal DNA, (i.e., between 50 and 700 bp as measured by Tapestation), remained at approximately 50%. Fig 3E represents the cfDNA profile from a different subject abstaining for either 1 day or 2 days. In this case, there was no difference in either the cfDNA profile or the yield. The image in Fig 3D and E also illustrate the often unique, but reproducible, cfDNA size-distribution profiles from the same subject.

## Methylation profile

The main objectives of this study were to determine the tissue of origin of the cfDNA present in seminal fluid and to understand whether the source of high molecular weight cfDNA was the same, or different, from that of the small (presumably nucleosomal) cfDNA. To do this, we performed size selection of the cfDNA, followed by enzymatic methylation analysis. To determine the relative contribution of different tissues to the cfDNA within a sample, we created a reference methylation matrix for tissues potentially contributing to seminal plasma, suitable for tissue deconvolution (S4 Table). Unsupervised clustering of the methylation markers for each reference tissue showed strong agreement within their tissue and mostly poor correlation across tissues (S4 Fig). Methylation values for each tissue set were plotted for the refence dataset alone (S5 Fig), reference dataset and SSD (S6 Fig), and reference dataset and HMW (S7 Fig). Using ENCODE datasets [19,20] (n = 102) from 7 different matched tissues, we found that, on average, 45.0% of methylation markers had a DNase I site from the marker's tissue within 250 bp, compared to an average of 3.7% for DNase I sites from other tissues (S5 Table), suggesting tissue functionality.

Fig 4A illustrates the tissue deconvolution results of the size selected small DNA (SSD) for 15 men without vasectomies (see S2 Fig. for results for men with vasectomies). In 13 of the 15 subjects, sperm was by far the most predominant signal contributing to over 70% of the total cfDNA

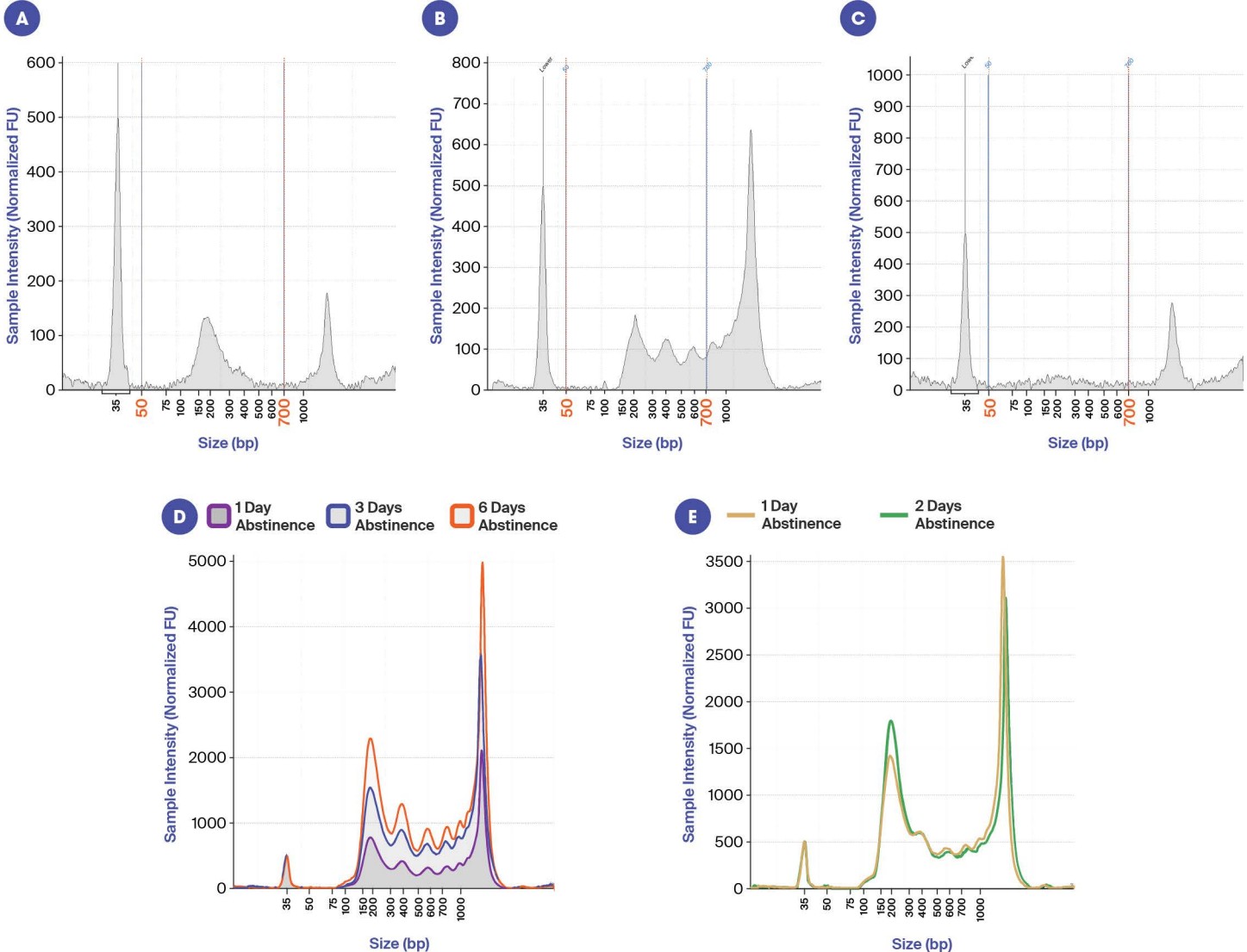

**Fig 3. Electropherogram images from various subjects illustrating the variability in size profiles and impact of abstinence time.** A: Example of a participant with two dominant cfDNA peaks, one nucleosomal cfDNA peak and one > 1 kb. B: Example of a participant with what appears to be multi-nucleosomal cfDNA peaks and one peak > 1 kb. C: Example of a vasectomy participant with just one peak > 1 kb. While these profiles were typically seen in subjects having undergone vasectomy, they were also seen occasionally in subjects not having undergone vasectomy. D: Three different seminal fluid cfDNA samples from the same participant after either 1, 3 or 6 days of abstinence. E: Two different seminal fluid cfDNA samples from the same patient after either 1 or 2 days of abstinence.

signal. The other 2 subjects had a very high granulocyte signal and less than 20% of the signal was from sperm. The overall predicted prostate fraction for SSD samples was high compared to blood [15], with a median of 2.7% and a maximum of 20%. Interestingly, the tissue deconvolution results from the size selected high molecular weight (HMW) cfDNA showed an increased proportion of somatic cell signal and decreased proportion of sperm signal (Fig 4B). The proportion of predicted prostate contribution reached as high as 26.5%, with a median of 10.9%, which is of notable significance for the field of liquid biopsy. The HMW DNA showed a marked reduction in sperm signal, with a median reduction of 28.7% compared to matched SSD samples (Fig 4C). Somatic tissues increased in relative predicted fraction in HMW vs SSD samples, with prostate (median + 6.2%), granulocyte (median + 4.9%), and kidney (median + 2.9%) showing

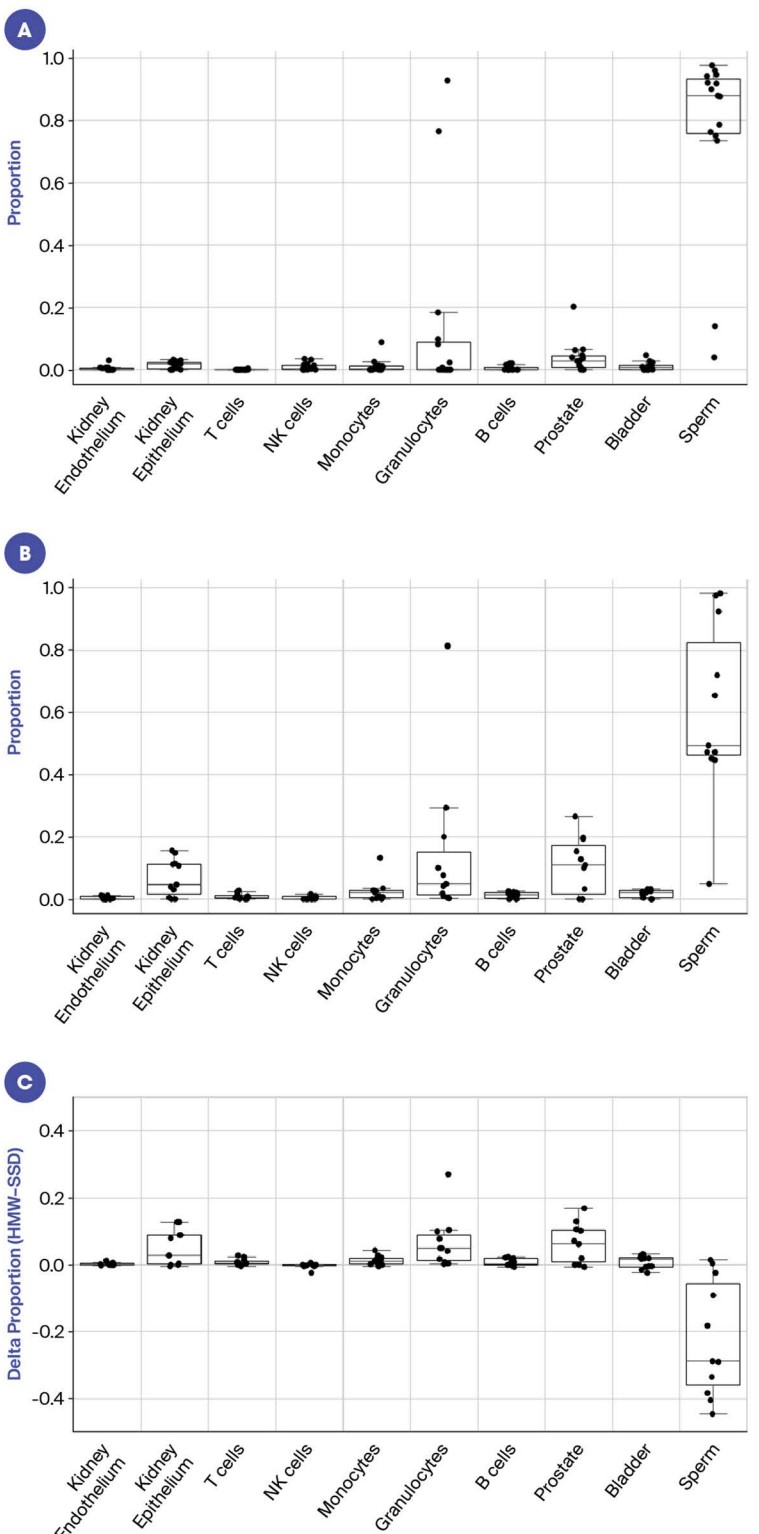

**Fig 4. Tissue deconvolution results illustrate HMW has lower proportion of sperm signal and higher proportion of somatic cell signal.** Panels A–C represent non-vasectomy subjects only. A: Tissue deconvolution results for SSD (n = 15). B: Tissue deconvolution results for HMW (n = 11). C: Difference in proportion of signal between HMW and SDD (n = 11).

predicted increased contribution. These findings demonstrate that the HMW fraction of cfDNA from seminal fluid harbors a higher abundance of somatic cell cfDNA than the nucleosomal fraction. Age did not seem to be a significant factor in the composition of cfDNA (S3 Fig). While there was a slight trend for lower sperm signal with increasing age, this was not significant (p-value = 0.16 for HMW and p-value =0.65 for SSD). The tissue deconvolution profile was shown to be similar with and without preservative for 3 different subjects on day 0. The profile was maintained in samples stored for up to 3 days in the oven with preservative (S8 Fig).

## Discussion

During apoptosis, DNA is fragmented at inter-nucleosomal linker sites and thus the resulting cfDNA is typically around 166 bp, which represents the size of the nucleosome DNA bound to the histone core (146 bp) and a (20 bp) linker connecting the nucleosomes [21]. However, cfDNA fragments from tumors and within the fetal fraction of pregnant women tend to be shorter [22]. In this study, the electropherograms indicated nucleosomal peaks closer to 200 bp rather than 166 bp, but upon sequencing SSD the mean peak insert size was 146 bp. We believe the insert size to be a more reliable marker of fragment size and believe the larger size seen on electropherogram to be some sort of artifact. Given that 85% of mature sperm DNA is associated with protamines as opposed to histones [23], it also begs the question of whether the sperm signal within the SSD fraction is coming from mature sperm or cells prior to spermiogenesis. Protamine associated DNA forms large toroidal structures of approximately 50,000 kb [24], therefore the majority of cfDNA from apoptotic mature sperm would not be expected to be in the SSD fraction. We also do not expect the HMW fraction to contain an abundance of protamine associated DNA since we deliberately performed a high speed centrifugation step to pellet the protamine toroidal structures. It was interesting to see various unique, but reproducible, cfDNA profiles between subjects. For example, some subjects had a single clean peak of mono-nucleosomal cfDNA, while others had very clear multi-nucleosomal peaks. The underlying biology of this warrants further investigation, but it could be related to the different amounts of DNases present in semen or the efficiency of cfDNA clearing, which could be of urological clinical significance. Our data also suggests that while increased abstinence times do not change the overall profile of cfDNA (or proportion of small to HWM cfDNA), the total amount of cfDNA increases as a function of abstinence time.

Previous studies have demonstrated that the yield of cfDNA obtained from seminal fluid is higher than that from other bodily fluids [25], and that fragment size and yield could potentially serve as a prostate cancer biomarker [5–7]. Ponti, et al [5–7] postulated that the high molecular weight DNA seen in prostate cancer patients was derived from necrotic prostate cancer cells [6]. However, the exact tissue source of seminal fluid cfDNA has not, until now, been determined. Here we demonstrate for the first time that in men aged 21–60 without vasectomy, most cfDNA usually arises from sperm (either mature or immature), but that there is an abundance of cfDNA that comes from various other cell types, including prostate, granulocytes and monocytes. In previous studies, researchers were careful to ensure no sperm lysis during sample collection and processing, and thus claimed that the DNA >1 kb was indeed "cell free" DNA as opposed to an artifact created during collection [25]. Another way to rule this out is to look at cfDNA profiles from men who have undergone vasectomy, thus eliminating the chance that sperm is lysed during collection and processing. In this study we observed the absence of a discrete peak of small, nucleosomal cfDNA in participants that had undergone vasectomy, but still a large amount of cfDNA >1 kb. This essentially proves that the larger cfDNA (>1 kb) is coming from cells, at least in part, other than sperm. It is worth noting however, that the majority of this cohort were under 40 yrs of age, and as such, unlikely

to have prostate cancer. It is therefore unlikely that in this cohort, the > 1 kb cfDNA is coming from necrotic prostate cancer cells.

Methylation analysis of cfDNA has emerged as a powerful tool for determining the tissue of origin (TOO) of cfDNA fragments, which has significant implications for diagnostics and monitoring of various diseases. To fully understand the TOO of cfDNA in seminal plasma, we performed a series of size selections to purify fragments < 500 bp (small cfDNA, SSD) and > 800 bp (HMW) and then performed methylation analysis on these two fractions. Our results demonstrate that in men having undergone a vasectomy there was no (significant) sperm signal in either fraction, and that in men not having undergone a vasectomy, the main source of cfDNA in both fractions was usually from sperm. In blood from healthy individuals, more than 90% of cfDNA is derived from white blood cells, with vascular endothelial cells and hepatocytes being the only detectable solid tissue source [15]. In this study we demonstrate that prostate derived cfDNA can be highly abundant in healthy controls demonstrating seminal fluid to be a potentially more useful sample type for studying disorders of the prostate. The most striking observation was that the somatic cell signal was usually higher in the HMW fraction than in the SSD fraction. This result is counterintuitive to the world of blood liquid biopsies, whereby the presence of HMW DNA is seen negatively in that it indicates a higher amount of background noise. From a diagnostics perspective, our data suggests that one could improve the detection sensitivity for various pathological conditions (for example, prostate cancer, prostatitis or benign prostate hyperplasia) by enriching the HMW portion of cfDNA in seminal fluid.

One limitation of our study is that we did not have access to reference datasets for all tissue/cell types present in the male reproductive tract (for example, epididymis or Sertoli cells) in building the deconvolution algorithm. Consequently, the proportion estimates of the various tissue/cell types are likely overestimated (given that the algorithm sums the signal to 1). This likely explains why in vasectomy subjects, the level of sperm was not zero. Also, given almost 78% of the testes is thought to consist of spermatogonial stem cells [26], which have almost identical methylation patterns as mature sperm [27], we are likely capturing cfDNA from these cells within the sperm signal. Another limitation is that the cohort of subjects included in this study was limited to males 21–60 yrs of age and limited medical history was obtained. To understand the potential utility of this approach for detection of prostate associated diseases such as cancer, prostatitis or benign prostate hyperplasia, further studies will be required using the relevant patient populations.

## Supporting information

**S1 Fig. Association of age with cfDNA yield and seminal fluid volume.** Solid circles are non-vasectomy subjects, open circles are vasectomy subjects. A: Normalized yield (ng/ml seminal plasma) versus age. B: Seminal fluid volume versus age.
(JPG)

**S2 Fig. Tissue deconvolution results for vasectomy subjects.** A: Tissue deconvolution results for size selected small cfDNA (SSD). B: Tissue deconvolution results for high molecular weight cfDNA (HMW). C: Difference in proportion of signal between HMW and SDD (n = 4). It is important to note that the algorithm used in the tissue deconvolution sums the signal to add to 1 (i.e.,100%), therefore while it appears that the vasectomy subjects have a lot more signal from the various somatic cells, this is driven by the absence of any sperm signal.
(JPG)

**S3 Fig. Tissue deconvolution by age for non-vasectomy subjects.** A: HMW tissue deconvolution by age. B: SSD tissue deconvolution by age.
(JPG)

**S4 Fig. Unsupervised clustering of reference dataset using our deconvolution signature matrix.** Pearson correlation followed by clustering of the n = 25 methylation markers for each reference tissue show strong agreement within their tissue and mostly poor correlation across tissues. An exception is the slight intermixing of the granulocyte/monocyte marker clusters, which can be explained by their similar developmental origins.
(JPG)

**S5 Fig. Methylation values for tissue/cell type specific hypomethylation markers in reference dataset.** Each graph represents methylation values for a tissue/cell type specific set of hypomethylation markers. Shown are methylation values for background (all other tissue/cell types) and the tissue/cell type of interest for all samples in the reference dataset.
(JPG)

**S6 Fig. Methylation values for tissue/cell type specific hypomethylation markers in size selected small cfDNA (SSD).** Each graph represents methylation values for a tissue/cell type specific set of hypomethylation markers. Shown are methylation values for background (reference dataset), tissue of interest (reference dataset), non-vasectomy seminal plasma (SP) samples and vasectomy SP samples. Note that the methylation profile for most of the non-vasectomy SP more closely matches sperm more so than any other tissue type, illustrating sperm to be the most prominent cell type. In vasectomy samples, prostate, granulocytes and monocytes markers appear hypomethylated compared to background tissues, suggesting the presence of DNA from these cell types within these samples.
(JPG)

**S7 Fig. Methylation values for tissue/cell type specific hypomethylation markers in high molecular weight cfDNA (HMW).** Each graph represents methylation values for a tissue/cell type specific set of hypomethylation markers. Shown are methylation values for background (reference dataset), tissue of interest (reference dataset), non-vasectomy seminal plasma (SP) samples and vasectomy SP samples. Note the very large variability in signal for sperm markers in non-vasectomy samples demonstrating a large range of sperm signal present.
(JPG)

**S8 Fig. Tissue deconvolution with and without preservative.** Graphs show tissue deconvolution results for 3 subjects at day 0 with and without preservative and at day 3 in the oven with preservative. A: HMW results: Subject 1 has predominantly granulocyte signal with some monocyte signal, while subjects 2 and 3 have predominantly sperm signal. On day 0 for all the 3 subjects the results look very similar whether preservative was added or not. Additionally, for all 3 subjects, the relative proportion of each cell type is similar on day 3 with preservative to day 0. B: SSD results: Subject 1 again has predominantly granulocyte signal with some monocyte signal, while subjects 2 and 3 have almost 100% sperm signal at day 0 with and without preservative and at day 3 with preservative.
(JPG)

**S1 Table. Reference datasets used for tissue deconvolution.**
(XLSX)

**S2 Table. Top selected tissue specific hypomethylation markers.**
(XLSX)

**S3 Table. ENCODE DNase I datasets used to check for overlap with tissue markers.**
(XLSX)

**S4 Table. Tissue methylation signature matrix.**
(XLSX)

**S5 Table. Overlap between tissue methylation markers and DNase I sites.**
(XLSX)

## Acknowledgments

We thank Mark Ferguson for his support in generating graphics, Daniel Colin and Awesta Froz for their contributions to cfDNA extraction, and William Mathews, Meredith Halks Miller, James Wingrove and Alice Chen for their thoughtful discussions. We also thank the study participants without whom this research would not have been possible.

## Author contributions

**Conceptualization:** Stephanie Huang, James C. Hart, Kim M. Clark-Langone.

**Formal analysis:** James C. Hart.

**Methodology:** Stephanie Huang, James C. Hart, Shellie Bench, Laura Rivas Yepes.

**Project administration:** James F. Smith, Bailey Griscom.

**Software:** James C. Hart.

**Supervision:** Stephanie Huang, Kim M. Clark-Langone.

**Writing – original draft:** Kim M. Clark-Langone.

**Writing – review & editing:** Stephanie Huang, James C. Hart, James F. Smith, Shellie Bench, Laura Rivas Yepes, Bailey Griscom.

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
