## [Decision Letter · Decision Letter 0]

24 Oct 2024

PONE-D-24-42704Tissue of origin characterization of cell free DNA in seminal plasma: Implications for new liquid biopsiesPLOS ONE

Dear Dr. Clark-Langone,

Thank you for submitting your manuscript to PLOS ONE. After careful consideration, we feel that it has merit but does not fully meet PLOS ONE’s publication criteria as it currently stands. Therefore, we invite you to submit a revised version of the manuscript that addresses the points raised during the review process.

 Please check the reviewers' comments, especially those regarding bias control, statistical multivariate analysis, and the transparency (open science) of your methods and results. Though data is accessible in a public repository, the methods should be offered in a code bank.

We look forward to receiving your revised manuscript.

Kind regards,

Alexis G. Murillo Carrasco

Academic Editor

PLOS ONE

“This study was funded by Fellow Health Inc.”

“Funding was provided by Fellow Health Inc. We thank Mark Ferguson for his support in generating graphics, Daniel Colin and Awesta Froz for their contributions to cfDNA extraction, and William Mathews, Meredith Halks Miller, James Wingrove and Alice Chen for their thoughtful discussions. We also thank the study participants without whom, this research would not have been possible.”

“This study was funded by Fellow Health Inc.”

4. We note that you have a patent relating to material pertinent to this article. Please provide an amended statement of Competing Interests to declare this patent (with details including name and number), along with any other relevant declarations relating to employment, consultancy, patents, products in development or modified products etc. Please confirm that this does not alter your adherence to all PLOS ONE policies on sharing data and materials, as detailed online in our guide for authors http://journals.plos.org/plosone/s/competing-interests by including the following statement: "This does not alter our adherence to  PLOS ONE policies on sharing data and materials.” If there are restrictions on sharing of data and/or materials, please state these. Please note that we cannot proceed with consideration of your article until this information has been declared.

Reviewers' comments:

Reviewer's Responses to Questions

**Comments to the Author**

1. Is the manuscript technically sound, and do the data support the conclusions?

Reviewer #1: Partly

Reviewer #2: Yes

Reviewer #3: Yes

2. Has the statistical analysis been performed appropriately and rigorously? 

Reviewer #1: No

Reviewer #2: Yes

Reviewer #3: Yes

3. Have the authors made all data underlying the findings in their manuscript fully available?

Reviewer #1: Yes

Reviewer #2: Yes

Reviewer #3: Yes

4. Is the manuscript presented in an intelligible fashion and written in standard English?

Reviewer #1: Yes

Reviewer #2: Yes

Reviewer #3: No

5. Review Comments to the Author

Reviewer #1: The present manuscript is a description of the potential sources of cfDNA in SP. Although it might be interesting for those in the field, the novelty of the study is limited. In addition, this reviewer has several methodological concerns. First, there were samples that were preserved and other were not. This might affect the results, and it is not something the authors seem to be considering in their subsequent analysis. Other factors that could be affecting the results are i) the age of the patients included, which ranges from 20-60 years old; and ii) the n of each group. In addition, the preservative composition used is not detailed at all and, therefore, the reproducibility of the results is compromised.

Reviewer #2: This study is an interesting analysis of seminal fluid DNA, showing DNA from other tissues in this material and suggesting that this type of samples seem to have the potential to be used as a liquid biopsies for the monitoring and identification of benign prostatic hyperplasia, prostatitis, and prostate cancer.

I have a few major comments:

The authors mention "The sequencing data will be available after acceptance on NIH BioProject.

https://www.ncbi.nlm.nih.gov/bioproject/PRJNA1162531" However, for code reproducibility it would be important for the authors to include their code on Github or a similar page, as well as a full analytical pipeline so their results can be reproduced.

It is important to validate identified prostate/kidney/granulocytes markers with other datasets. DNAm is a key marker of tissue identity that tends to overlap with DNase I hotspots (another key marker of tissue identity) for the tissue of origin. To confirm that the identified sites are genuinely from the tissues listed (prostate, etc.) and have biological roles in these tissues it would be important to show preferential overlap with DNase I hotspots for prostate/kidney/granulocytes. This could be done using eFORGE (https://eforge.altiusinstitute.org/, there are probably a range of sites that overlap between the DNAm markers identified here and the EPIC array), or the eFORGE table S3 and S4 panel of blood and kidney tissue-specific differentially methylated positions (tDMPs). Other analyses using other chromatin datasets such as histone marks or chromatin states could also help clarify this issue.

Table S2 seems to be missing statistics/p-value and other information used in ranking top DNA methylation (DNAm) positions.

Reviewer #3: Abstract Pg 1 not clear

While sperm was the predominant signal in most men without vasectomies, the proportion of predicted prostate contribution reached as high as 26.5% in the HMW fraction. In other subjects without vasectomies, granulocyte cfDNA made up most of the signal.

What is the preservative?

What is the significance of using preservative immediately and leaving the sample for three days?

The english of the manuscript needs to be more lucid.

---

## [Author Response · Author response to Decision Letter 1]

4 Nov 2024

Dear Dr. Alexis G. Murillo Carrasco,

Thank you for your response regarding our manuscript. We have gone through the style requirements guide and have updated the manuscript accordingly, including updating the Refences with the correct style. We have also removed the funding source from the acknowledgements section. The funder, Fellow Health Inc., is the employer of all authors, and funded salaries, sample collection, reagents, instrumentation and sequencing costs.

We also appreciate the comments received from the 3 reviews. Below is a description of the changes made to address each point raised by them.

Reviewer #1: First, there were samples that were preserved and other were not. This might affect the results, and it is not something the authors seem to be considering in their subsequent analysis.

Response: We have now included Figure S8 which shows the tissue deconvolution results for 3 subjects with and without preservative on day 0. Hopefully this addresses the reviewer’s concern and illustrates that tissue deconvolution results for the same subject are similar on day 0 with and without preservative. Figure S8 also shows that with preservative on day 3 in the oven, the tissue deconvolution results still look similar to day 0. One would not expect the results to look the same on day 3 without preservative due to cell lysis; the Tapestation images clearly show that there is a much higher yield of DNA after 3 days in the oven when there is no preservative.

Reviewer #1: Other factors that could be affecting the results are i) the age of the patients included, which ranges from 20-60 years old; and ii) the n of each group.

Response: We have added additional graphs to the supplementary information regarding patient age. Figure S1 shows normalized cfDNA yield and total semen volume by age. Figure S3 shows tissue deconvolution results by age. The number within each age category can clearly be seen in both S1 and S3.

Reviewer #1: In addition, the preservative composition used is not detailed at all and, therefore, the reproducibility of the results is compromised.

Response: The preservative is proprietary to Fellow Health Inc. who funded this work, so we are unable to provide the composition. However, we would be willing to provide others with the preservative for reproduction of the results. Additionally, since we have illustrated that the tissue deconvolution results are the same with and without preservative on day 0, others could simply look at tissue deconvolution without preservative on day 0.

Reviewer #2: The authors mention "The sequencing data will be available after acceptance on NIH BioProject.

https://www.ncbi.nlm.nih.gov/bioproject/PRJNA1162531" However, for code reproducibility it would be important for the authors to include their code on Github or a similar page, as well as a full analytical pipeline so their results can be reproduced.

Response: Scripts needed to run the methylation tissue deconvolution have been uploaded to github at https://github.com/meetfellow/fellow-genomics. This link has also been added to the manuscript under the methylation tissue deconvolution methods section.

Reviewer #2: It is important to validate identified prostate/kidney/granulocytes markers with other datasets. DNAm is a key marker of tissue identity that tends to overlap with DNase I hotspots (another key marker of tissue identity) for the tissue of origin. To confirm that the identified sites are genuinely from the tissues listed (prostate, etc.) and have biological roles in these tissues it would be important to show preferential overlap with DNase I hotspots for prostate/kidney/granulocytes. This could be done using eFORGE (https://eforge.altiusinstitute.org/, there are probably a range of sites that overlap between the DNAm markers identified here and the EPIC array), or the eFORGE table S3 and S4 panel of blood and kidney tissue-specific differentially methylated positions (tDMPs). Other analyses using other chromatin datasets such as histone marks or chromatin states could also help clarify this issue.

Response: We validated the potential biological role of these tissue specific methylation markers using ENCODE DNase I hypersensitivity data. Using n=102 ENCODE datasets from 7 different matched tissues, we looked for proximity between the methylation markers and DNase I hypersensitive sites. We found that, on average, 45.0% of methylation hypo markers had a DNase I site from the marker’s tissue within 250bp, compared to an average of 3.7% for non-tissue DNase I sites. These results are now illustrated in Table S3 and discussed in the results section.

Reviewer #2: Table S2 seems to be missing statistics/p-value and other information used in ranking top DNA methylation (DNAm) positions.

Response: We have added Table S2 to address this comment. For each marker it shows the tissue it marks, as well as its separation score (described in methods) that we used to rank markers and select the top 25.

Reviewer #3: Abstract Pg 1 not clear

While sperm was the predominant signal in most men without vasectomies, the proportion of predicted prostate contribution reached as high as 26.5% in the HMW fraction. In other subjects without vasectomies, granulocyte cfDNA made up most of the signal.

Response: We have modified as follows: “While sperm was the predominant signal in most men without vasectomies, granulocyte cfDNA made up most of the signal in two of the non-vasectomy subjects. Unexpectedly, the proportion of prostate signal reached as high as 26.5% in the HMW fraction in non-vasectomy subjects”.

Reviewer #3: What is the preservative?

Response: The preservative is proprietary to Fellow Health Inc. who funded this work, so we are unable to provide the composition. However, we would be willing to provide others with the preservative for reproduction of the results. Additionally, since we have illustrated that the tissue deconvolution results are the same with and without preservative on day 0, others could simply look at tissue deconvolution without preservative on day 0.

Reviewer #3: What is the significance of using preservative immediately and leaving the sample for three days?

Response: If a test developer were to want to make an at home collection kit, they would need to ensure that the sample is stable during shipping. The significance of adding the preservative immediately and leaving it for 3 days was to demonstrate that the preservative can stabilize cfDNA (and prevent sperm lysis) if shipping back to the laboratory were to take 3 days. The following sentence was added to the manuscript, “Adding preservative immediately and waiting for 3 days before preparing seminal plasma best represents the scenario of an at home collection test.”

Reviewer #3: The english of the manuscript needs to be more lucid.

Response: We have modified the part of the abstract that was not clear to reviewer #3 to the following: “While sperm was the predominant signal in most men without vasectomies, granulocyte cfDNA made up most of the signal in two of the non-vasectomy subjects. Unexpectedly, the proportion of prostate signal reached as high as 26.5% in the HMW fraction in non-vasectomy subjects”. If there are other sentences or paragraphs that are not clear we can certainly re-write. The other two reviewers answered, “yes” to the question, “Is the manuscript presented in an intelligible fashion and written in standard English?” so we are not sure which parts of the manuscript are not “lucid”.

We also made some additional changes to Figures as summarized below:

1. Figure 3: For panels D and E we changed to color images as it was difficult to see the various different samples in grayscale.

2. Figure 4: For panel C, we reduced the range on the y-axis so that changes were more obvious (there was a lot of white space in the previous plot).

3. Figure S1, that is now S2: For panel C, we reduced the range on the y-axis so that changes were more obvious (there was a lot of white space in the previous plot).

Finally, we also corrected the subject and library numbers in the Materials and Methods section for methylation to match the correctly annotated Figure 4, its associated legend, and the results section. There were actually 15 SSD non-vasectomy libraries (not 14) and all 15 HMW samples had matching SSD samples.

We hope that with these changes, the manuscript is now suitable for acceptance.

Best,

Kim Clark-Langone

---

## [Editor Report · Decision Letter 1]

7 Nov 2024

PONE-D-24-42704R1Tissue of origin characterization of cell free DNA in seminal plasma: Implications for new liquid biopsiesPLOS ONE

Dear Dr. Clark-Langone,

Thank you for submitting your manuscript to PLOS ONE. After careful consideration, we feel that it has merit but does not fully meet PLOS ONE’s publication criteria as it currently stands. Therefore, we invite you to submit a revised version of the manuscript that addresses the points raised during the review process.

 Before sending this new version of your manuscript to reviewers, I want to check some open science features. Though you shared your code, the BioProject data is blocked until the article is published. Could you please provide a reviewer link to facilitate reviewers' access until publication? You can find more information about this reviewer's link here (https://www.ncbi.nlm.nih.gov/sra/docs/submitquestions/). Please submit your revised manuscript by Dec 22 2024 11:59PM. If you will need more time than this to complete your revisions, please reply to this message or contact the journal office at plosone@plos.org . Please include the following items when submitting your revised manuscript:

We look forward to receiving your revised manuscript.

Kind regards,

Alexis G. Murillo Carrasco

Academic Editor

PLOS ONE

---

## [Author Response · Author response to Decision Letter 2]

8 Nov 2024

In response to the reviewer's comment, "Though you shared your code, the BioProject data is blocked until the article is published." The data should no longer be blocked and should now be accessible. The reviewer can also use this link to directly access the data:

https://www.ncbi.nlm.nih.gov/Traces/study/?acc=PRJNA1162531&o=acc_s%3Aa

---

## [Decision Letter · Decision Letter 2]

4 Dec 2024

PONE-D-24-42704R2Tissue of origin characterization of cell free DNA in seminal plasma: Implications for new liquid biopsiesPLOS ONE

Dear Dr. Clark-Langone,

Thank you for submitting your manuscript to PLOS ONE. After careful consideration, we feel that it has merit but does not fully meet PLOS ONE’s publication criteria as it currently stands. Therefore, we invite you to submit a revised version of the manuscript that addresses the points raised during the review process.

 Please check the reviewer's comments and modify the manuscript accordingly. In particular, check the results that must be adjusted for confounding variables. Please be aware that this adjustment could influence the interpretation of results. Therefore, modify the result descriptions if required.

We look forward to receiving your revised manuscript.

Kind regards,

Alexis G. Murillo Carrasco

Academic Editor

PLOS ONE

Journal Requirements:

Reviewers' comments:

Reviewer's Responses to Questions

**Comments to the Author**

1. If the authors have adequately addressed your comments raised in a previous round of review and you feel that this manuscript is now acceptable for publication, you may indicate that here to bypass the “Comments to the Author” section, enter your conflict of interest statement in the “Confidential to Editor” section, and submit your "Accept" recommendation.

Reviewer #2: (No Response)

Reviewer #3: All comments have been addressed

2. Is the manuscript technically sound, and do the data support the conclusions?

Reviewer #2: Yes

Reviewer #3: Yes

3. Has the statistical analysis been performed appropriately and rigorously? 

Reviewer #2: Yes

Reviewer #3: I Don't Know

4. Have the authors made all data underlying the findings in their manuscript fully available?

Reviewer #2: Yes

Reviewer #3: Yes

5. Is the manuscript presented in an intelligible fashion and written in standard English?

Reviewer #2: Yes

Reviewer #3: Yes

6. Review Comments to the Author

Reviewer #2: Re "We found that, on average, 45.0% of methylation hypo markers had a DNase I site from the marker’s tissue within 250bp, compared to an average of 3.7% for non-tissue DNase I sites. These results are now illustrated in Table S3 and discussed in the results section." These results are encouraging but there are several sources of confounding that can be driving this including promoter content and CpG content/CpG Island density (hence the suggestion of using eFORGE). Can you adjust for these confounders or show that even a subset of CpGs are still enriched after adjusting for promoter content or CpG content/CpG Island density (with code on GitHub)?

Please ensure that vectorial figures are included (vectorial and not bitmap) so they are readable in the PLOS ONE pdf. Current figures are grainy and including vectorial (or high resolution) images will greatly improve this work.

Reviewer #3: All the queries have been taken care of and the changes are properly done in the manuscript, so, looks good now.

7. PLOS authors have the option to publish the peer review history of their article (what does this mean? ). If published, this will include your full peer review and any attached files.

**Do you want your identity to be public for this peer review?** For information about this choice, including consent withdrawal, please see our Privacy Policy .

Reviewer #2: No

Reviewer #3: **Yes: ** Dr. Sudipta Saha

---

## [Author Response · Author response to Decision Letter 3]

9 Dec 2024

Thank you for your response regarding our manuscript. We also appreciate the additional comments received from reviewer #2. Please see below for our response to each comment.

Reviewer #2: Re "We found that, on average, 45.0% of methylation hypo markers had a DNase I site from the marker’s tissue within 250bp, compared to an average of 3.7% for non-tissue DNase I sites. These results are now illustrated in Table S3 and discussed in the results section." These results are encouraging but there are several sources of confounding that can be driving this including promoter content and CpG content/CpG Island density (hence the suggestion of using eFORGE). Can you adjust for these confounders or show that even a subset of CpGs are still enriched after adjusting for promoter content or CpG content/CpG Island density (with code on GitHub)?

Response: One of the major objectives of this study was to develop a tissue deconvolution approach that could be used with seminal plasma. Our methodology was based a Nature paper (Loyfer et al, 2023) whereby they determined potential tissue-specific biomarkers for use in liquid biopsies. We supplemented these datasets with data sourced from other studies to capture the full expected tissue content of seminal plasma (i.e. sperm). We realize that eFORGE can be useful in understanding tissue-specific relevance and potential functional roles of hypomethylated sites, but this was not one of the aims of this study. The studies that were the source of our reference methylation data have already explored much of the potential functional nature of such marks. We undertook a good faith analysis to satisfy the request of reviewer #2 and feel that further analysis is unwarranted as the relationship between DNase I sensitivity and DNA methylation is complex and context-dependent. We do not feel that the use of eFORGE would be suitable for our work, as this database has few overlapping tissues with our reference methylation data. It also uses methylation arrays, which have low overlap with our methylation enrichment panel, and as such we do not believe it appropriate for the suggested analysis. We also note that our methylation panel was not restricted to promoter regions but included a wide range of diverse genomic locations.

We have modified our updated language (in response to the original comment by reviewer #2) to remove the notion that we were “verifying” or “validating” tissue functionality as part of this work and are happy to completely remove any language relating to functionality given this was not an aim of this study. Again, the main objective was to establish a methodology for tissue deconvolution based on the Nature paper by Loyfer et al, 2023, not to demonstrate tissue functionality.

Reviewer #2: Please ensure that vectorial figures are included (vectorial and not bitmap) so they are readable in the PLOS ONE pdf. Current figures are grainy and including vectorial (or high resolution) images will greatly improve this work.

Response: We have modified images to meet PLOS One requirements.

---

## [Decision Letter · Decision Letter 3]

3 Jan 2025

Tissue of origin characterization of cell free DNA in seminal plasma: Implications for new liquid biopsies

PONE-D-24-42704R3

Dear Dr. Clark-Langone,

We’re pleased to inform you that your manuscript has been judged scientifically suitable for publication and will be formally accepted for publication once it meets all outstanding technical requirements.

Kind regards,

Alexis G. Murillo Carrasco

Academic Editor

PLOS ONE

Additional Editor Comments (optional):

Reviewers' comments:

Reviewer's Responses to Questions

**Comments to the Author**

1. If the authors have adequately addressed your comments raised in a previous round of review and you feel that this manuscript is now acceptable for publication, you may indicate that here to bypass the “Comments to the Author” section, enter your conflict of interest statement in the “Confidential to Editor” section, and submit your "Accept" recommendation.

Reviewer #3: All comments have been addressed

2. Is the manuscript technically sound, and do the data support the conclusions?

Reviewer #3: Yes

3. Has the statistical analysis been performed appropriately and rigorously? 

Reviewer #3: I Don't Know

4. Have the authors made all data underlying the findings in their manuscript fully available?

Reviewer #3: Yes

5. Is the manuscript presented in an intelligible fashion and written in standard English?

Reviewer #3: Yes

6. Review Comments to the Author

Reviewer #3: The manuscript "Tissue of origin characterization of cell free DNA in seminal plasma: Implications for

new liquid biopsies" looks better now.

---

## [Editor Report · Acceptance letter]

PONE-D-24-42704R3

PLOS ONE

Dear Dr. Clark-Langone,

I'm pleased to inform you that your manuscript has been deemed suitable for publication in PLOS ONE. Congratulations! Your manuscript is now being handed over to our production team.

Kind regards,

on behalf of

Dr. Alexis G. Murillo Carrasco

Academic Editor

PLOS ONE